# The Association between Caffeine Consumption from Coffee and Tea and Sleep Health in Male and Female Older Adults: A Cross-Sectional Study

**DOI:** 10.3390/nu16010131

**Published:** 2023-12-30

**Authors:** Mette van der Linden, Margreet R. Olthof, Hanneke A. H. Wijnhoven

**Affiliations:** 1Department of Health Sciences, Faculty of Science, Vrije Universiteit Amsterdam, 1081 HV Amsterdam, The Netherlands; 2Department of Health Sciences, Faculty of Science, Amsterdam Public Health Research Institute, Vrije Universiteit Amsterdam, 1081 HV Amsterdam, The Netherlands

**Keywords:** older adults, sleep health, caffeine, sex differences, observational study

## Abstract

Poor sleep health is common in older adults and is associated with negative health outcomes. However, the relationship between caffeine consumption and sleep health at an older age is poorly understood. This study investigated the association between caffeine consumption and sleep health in community-dwelling older males and females in The Netherlands. Cross-sectional analyses were performed using data from 1256 participants aged 61–101 years from the Longitudinal Ageing Study Amsterdam. Self-reported questions assessed sleep disturbances (including sleep latency, continuity, and early awakening), sleep duration, and perceived sleep quality. Caffeine consumption was determined with questions about frequency, quantity, and type of coffee and tea consumption. Logistic and linear regression models were used, controlling for potential confounders, and interaction by sex and age was tested. Caffeine consumption showed significant interactions with sex (*p* < 0.005) in association with sleep health outcomes. Older females who abstained from caffeine consumption reported more sleep disturbances (β = 0.64 [95%CI 0.13; 1.15]) and had greater odds of short sleep duration (<7 h/day) (OR = 2.26 [95% CI 1.22; 4.20]) compared to those who consumed caffeine. No associations were observed for long sleep duration (>8 h/day) and perceived sleep quality. No associations were observed in older males. Caffeine abstinence was associated with more sleep disturbances and short sleep duration in older females, but not in males. The observed association in older females may reflect reverse causation, suggesting that females may have different motivations for discontinuing caffeine consumption than males.

## 1. Introduction

Good sleep health is fundamental to human functioning, health, and well-being [1,2]. Its importance becomes even more evident during ageing as the prevalence of sleep disturbances and disorders increases with age [3,4,5]. Approximately one-fourth of older adults (≥65 years) do not meet the recommended sleep duration of 7 to 8 h per night, with symptoms of insomnia such as difficulty with sleep onset, sleep maintenance, and early morning awakening being particularly common [6]. Poor sleep quality and inadequate sleep duration in older adults are associated with adverse physical and mental health outcomes, including impaired cognitive function [7,8,9,10,11,12,13], reduced functional capacity [14,15,16], disability [17,18], reduced quality of life [19], and increased risk of all-cause mortality [10,20,21,22]. The causes of poor sleep health in older adults are multifactorial and can largely be attributed to age-related factors, including a higher prevalence of medical and psychiatric conditions, the use of multiple medications, and psychosocial or behavioural factors such as loneliness and depression [23]. In addition, nutrition—a modifiable lifestyle factor—may play a role [24].

Caffeine is a commonly used psychoactive substance found in a variety of foods and beverages [25]. In Europe, including The Netherlands, coffee and tea are the main sources of caffeine for adults [26,27]. While caffeine consumption has been associated with beneficial health outcomes, including a reduced risk of all-cause mortality, cardiovascular disease, and type 2 diabetes [28,29], it may also adversely affect sleep health. Caffeine is known to promote wakefulness by antagonizing adenosine A1 and A2 receptors in the brain [30,31]. Adenosine is a naturally occurring chemical in the human body that plays an important role in sleep regulation, primarily through its sleep-promoting effects when binding to its receptors [32]. Additionally, caffeine from coffee may also disrupt sleep by reducing the secretion of melatonin, a hormone responsible for regulating sleep patterns [33]. Controlled laboratory experiments and observational studies in adults have indicated that caffeine consumption can prolong sleep latency (the time it takes to fall asleep after going to bed) and reduce sleep duration, efficiency (defined as the ratio of total sleep time to time spent in bed), and perceived sleep quality [34,35]. However, few studies have examined the relationship between caffeine consumption and sleep health in older adults [36,37,38,39,40], despite the high prevalence of sleep disturbances and disorders in this population.

Previous laboratory experiments have shown that caffeine has a greater acute effect on sleep quality and duration in middle-aged adults compared to younger adults [41,42], particularly at a higher doses of 400 mg [41]. In addition, a large cross-sectional study found that self-reported insomnia attributed to caffeine consumption increased with age [43], suggesting that older adults may be more sensitive to the effects of caffeine than younger adults. However, the results of studies in older adults are inconsistent and inconclusive. Two cross-sectional studies found an association between higher caffeine consumption and shorter self-reported sleep duration [36,37]. In contrast, another cross-sectional study of 337 middle-aged and older adults found no association between caffeine consumption and sleep duration as measured using actigraphy [38]. The association between caffeine consumption and sleep quality in older adults is also inconsistent. One cross-sectional study of hospitalized older adults found that caffeine abstinence was associated with poorer sleep quality [39], whereas another cross-sectional study found that caffeine consumption was associated with better sleep quality in community-dwelling older adults and with poorer sleep quality in hospitalized older adults [40]. Another study of 428 older females found no association between caffeine consumption and sleep quality [37].

The present study aims to investigate the association between caffeine consumption from coffee and tea and sleep health in a representative sample of community-dwelling Dutch older adults, using data from the Longitudinal Ageing Study Amsterdam (LASA). In addition, this study aims to investigate whether associations differ by sex. The importance of considering sex differences in health research is increasingly recognized [44,45]. However, sex differences in the association between caffeine consumption and sleep have received little attention.

## 2. Materials and Methods

### 2.1. Design and Participants

For the current study, existing data from LASA were used. LASA is an ongoing cohort study of older adults in The Netherlands that focuses on the determinants, trajectories, and consequences of physical, cognitive, emotional, and social functioning [46]. Data were first collected in 1992–1993 from a cohort of 3107 participants aged 55 to 85 years. Follow-up measurement waves were scheduled roughly every three years thereafter. The sample was derived from a variety of municipalities across three culturally distinct regions in The Netherlands, making it nationally representative. From the same sampling frame, an additional second and third cohort of respondents aged 55 to 64 years were added to the original sample in 2002–2003 and 2012–2013, respectively. More information on sampling and data collection procedures can be found elsewhere [46,47,48]. For this cross-sectional study, we used data collected in 2018–2019 of 1701 participants aged 61 to 101 years from the first, second and third cohort. Data collection consisted of a structured main interview, a structured medical interview with clinical measurements, and a self-administered questionnaire. Participants who did not complete the self-administered questionnaire or whose data on caffeine consumption were missing or invalid were excluded from the analysis. All included participants provided written informed consent. The LASA study protocol was conducted in accordance with the Declaration of Helsinki and received approval from the medical ethics committee of the VU University Medical Centre (IRB numbers: 92/138, 2002/141, 2012/361, and 2016.301).

### 2.2. Sleep Disturbances, Sleep Duration, and Perceived Sleep Quality

A self-administered questionnaire assessed participants’ sleep health. The presence and frequency of sleep disturbances was measured using three categorical questions about having difficulties with sleep onset, sleep continuity, and early morning awakening. Questions were formulated as follows: ‘Do you experience difficulties falling asleep?’, ‘Do you experience interruptions in your sleep?’, and ‘Do you wake up too early?’. Response options to each of the three questions were ‘almost never’, ‘sometimes’, ‘frequently’, and ‘almost always’. Response options were assigned scores of one, two, three, and four, respectively. These scores were summed to compute a scale ranging from 3 (no sleep disturbances) to 12 (many sleep disturbances). A scale was only computed if participants answered all three questions. Sleep duration was measured with a single open-ended question about the number of hours participants usually sleep each night, registered as total minutes of sleep within 24 h. According to the recommend sleep duration for adults aged ≥ 65 years [49], the variable was categorized into three groups: recommended sleep duration (7–8 h/day), short sleep duration (<7 h/day), and long sleep duration (>8 h/day). Lastly, perceived sleep quality was measured with the question: ‘When you think about the past month, how would you rate your sleep quality?’ (good; somewhat good; poor; very poor). The variable was dichotomized into ‘good’ (good; somewhat good) and ‘poor’ (somewhat poor; very poor).

### 2.3. Caffeine Consumption from Coffee and Tea

The self-reported questionnaire assessed participants’ coffee and tea consumption over the past month. Participants were asked about their weekly coffee and tea consumption, specifying the frequency in days per week (none; <1; 1; 2; 3; 4; 5; 6; 7 days per week) and the quantity in cups per day (1; 2; 3; 4; 5; 6; 7; 8; 9; ≥10 cups) for each type of coffee (caffeinated; decaffeinated) and tea (black; green; rooibos; herbal; other namely …) on days they consumed these beverages. Teas reported under ‘other, namely …’ were subdivided into the existing response categories if appropriate. Furthermore, cup size was asked separately for coffee and tea. Response options were ‘small (about 125 mL)’, ‘medium (about 165 mL)’, ‘large (about 225 mL)’, ‘other, namely … mL’, and ‘don’t know’. In case of ‘other, namely … mL’, participants could provide a numerical response.

Average daily caffeine consumption from coffee and tea, in milligrams per day, was calculated for each participant as follows. First, the total volume in millilitres on days of consumption was calculated for caffeinated coffee, black tea, and green tea separately. This involved multiplying the reported number of cups a day by the corresponding cup size. If cup size was unknown (don’t know) or missing, this value was imputed by the mean cup volumes within the sample—160 mL for coffee and 178 mL for tea. Additionally, if a participant reported consuming 10 or more cups a day, a total of 10 cups was attributed. Subsequently, an average daily consumption in millilitres was calculated for each beverage by multiplying the total daily volume by the weekly consumption (frequency in days per week) and dividing it by seven (days). For participants indicating a consumption frequency of less than one day per week, a value of 0.5 days per week was assigned. To ascertain the average daily caffeine consumption in milligrams per day, the average daily consumption in millilitres for each beverage was multiplied by the respective caffeine content per 100 mL: 44.5 mg for coffee, 22.0 mg for black tea, and 15.1 mg for green tea [26]. Finally, the total daily caffeine consumption from both coffee and tea was calculated by summing caffeine consumption from each source per participant. As an example, if a participant reported drinking coffee five days a week, with a daily consumption of 2 cups of 125 mL, the calculation is as follows: ((2 cups × 125 mL) × 5 days)/7 days = 179 mL coffee on average per day. This corresponds to an average caffeine intake of 80 mg per day, calculated as follows: 179 mL × (44.5 mg/100 mL). In cases where a participant lacked information on caffeine consumption from one source (coffee or tea), while information on caffeine consumption from the other source was present, missing values were recoded to reflect zero caffeine consumption from that specific missing source (*n* = 161). For the analyses, caffeine consumption was divided into five categories. The first category included participants who abstained from caffeine consumption (0 mg/day). The subsequent four categories were defined based on quartiles for participants who reported caffeine consumption (>0 mg/day), yielding the following categories: low (≤173 mg/day), moderate (174–272 mg/day), high (273–367 mg/day), and very high (>367 mg/day).

### 2.4. Other Measurements

Potential sociodemographic (age, sex, level of attained education, partner status, and level of urbanization), health-related (number of chronic diseases, depressive symptoms, body mass index (BMI), and subjective pain), and behavioural (smoking status, alcohol use, and physical activity) confounders were selected a priori based on data availability, prior research, and reported associations with both exposure and outcome. During the main interview, participants were questioned about their sex at birth, age, and level of attained education. Response categories for level of attained education were combined into three categories: low (elementary not completed; elementary education; lower vocational education), intermediate (general intermediate education; intermediate vocational education; general secondary education), and high (higher vocational education; college education; university education). The level of urbanization—defined as the average number of addresses per square kilometre within a one-kilometre radius circle [50]—was measured using a zip code classification system designed by Statistics Netherlands (CBS, Heerlen/Voorburg, The Netherlands), which classifies zip codes into five urbanization levels: not (<500), little (500–1000), somewhat (1000–1500), highly (1500–2500), and very highly (≥2500). These groups were categorized into ‘sparsely populated (<1000)’ and ‘densely populated (≥1000)’. Information on partner status was self-administered and categorized as ‘living alone’ and ‘living with partner’. Subjective pain was assessed using a subscale based on the Nottingham Health Profile [51]. Participants were questioned about experiencing constant pain and experiencing pain when changing position, sitting, or walking. Data on health-related factors, alcohol consumption, and cigarette smoking were accumulated during the medical interview. Participants were questioned about the presence of chronic nonspecific lung disease, cardiac disease, peripheral arterial disease, stroke, diabetes mellitus, rheumatoid arthritis, and malignancies. Additionally, participants could report a maximum of two other chronic diseases for which symptoms or treatment had been present for at least three months. Depressive symptoms were measured with the Centre for Epidemiologic studies Depression Scale (CES-D) [52]. BMI was calculated as weight (kilograms)/height (meters)^2^. Weight was measured using a calibrated bathroom scale, and height was measured using a stadiometer. In the absence of measured weight or height, self-reported values were obtained. To define alcohol use, a standard developed by the Netherlands Economic Institute (NEI) was used, which categorizes alcohol use into four groups (no use, moderate use, grey area, and excessive use), corrected for sex [53]. The ‘excessive use’ group was merged with the ‘grey area’ group, creating a combined category labelled as ‘above moderate use’. Based on questions about current (yes/no) and past smoking (yes/no), participants were categorized as never smoked, former smoker, or current smoker. Physical activity was measured using the validated LASA Physical Activity Questionnaire (LAPAQ), an interviewer-administered questionnaire that estimates the frequency and duration of participation in activities over the past 2 weeks [54]. All activities in the LAPAQ were assigned a MET score based on previously published MET score lists [55,56] and interviews with activity experts. From this, a total amount of MET hours per week was calculated.

### 2.5. Statistical Analyses

Statistical analyses were performed using SPSS Statistics (version 28, IBM Corp., Armonk, NY, USA). A total of 445 participants were excluded from the analyses due to the non-completion of the self-administered questionnaire (*n* = 422) or because of missing or invalid data on caffeine consumption (*n* = 23), resulting in a final analytical sample of 1256 participants. Sample characteristics of included and excluded eligible participants were quantified using descriptive statistics. Continuous variables were presented as means with standard deviations (SD) if normally distributed. If not normally distributed, variables were presented as medians with interquartile ranges (IQR). Categorical variables were presented as proportions. Linear regression analyses were performed to assess the association between caffeine consumption and sleep disturbances. Multinominal logistic regression analyses were performed to assess the association between caffeine consumption and short (<7 h) and long (>8 h) sleep duration (with recommended sleep duration (7–8 h) as the reference group), and logistic regression analyses were performed to assess the association between caffeine consumption and perceived sleep quality. In each model, ‘low caffeine consumption (≤173 mg/day)’ was set as the reference group. Analyses were also performed with a binary caffeine classification: no caffeine consumption (0 mg/day) and caffeine consumption (>0 mg/day), with ‘caffeine consumption (>0 mg/day)’ set as the reference group. Both crude and adjusted models were fitted to account for potential confounding variables. Potential confounders were adjusted for by adding them to the regression models. Effect modification by sex and age was tested by adding interaction terms to the crude regression models and evaluating statistical significance of the interaction term (*p* < 0.05). If effect modification was present, stratified results were presented; otherwise, variables were included as potential confounders. To preserve sample size and increase the precision of the estimates, all covariates with missing values were imputed using the multiple imputation procedure in SPSS Statistics. This was performed despite all confounders having less than 5% missing values. A total of five datasets were imputed. Variables included in the imputation procedure were age, sex, level of attained education, partner status, level of urbanization, number of chronic diseases, depressive symptoms, BMI, subjective pain, smoking status, alcohol use, and physical activity.

## 3. Results

### 3.1. Sample Characteristics

A flowchart of the participants included in the analytical samples is shown in Figure 1. Compared to included participants, excluded eligible participants (*n* = 445) were somewhat older (mean age of 76 years versus 73 years), more often female (58% versus 53%), lower educated (45% versus 31% lower education), and had more chronic diseases (mean of 2.8 versus 2.2) (Table 1).

Sample characteristics of included participants (*n* = 1256) are presented in Table 1, stratified by sex and caffeine consumption (yes/no). The sample consisted of 587 (46.7%) males and 669 females. Seventeen percent of males and 26.3% of females had short sleep duration (<7 h/night), 25.7% of males and 43.4% of females experienced several to many sleep disturbances, and 12.8% of males and 22.5% of females reported poor perceived sleep quality. Compared to males, females had a lower mean caffeine intake (244 mg/day versus 286 mg/day in males), consumed less caffeine from coffee (median of 167 mg/day versus 220 mg/day in males) and more caffeine from tea (median of 50 mg/day versus 26 mg/day in males). The proportion of females who did not consume caffeine (9.1%) exceeded that of males (6.6%). Males and females who did not consume caffeine were on average older, lower educated, had more chronic diseases and depressive symptoms, and were physically less active.

### 3.2. Association between Caffeine Consumption and Sleep Health Outcomes

Statistically significant interactions were observed between sex and caffeine consumption in the association with sleep disturbances (*p* ≤ 0.002), short sleep duration (*p* ≤ 0.001), and perceived sleep quality (*p* = 0.005). No interactions were found for age. In both males and females, there was no association between the categories of caffeine consumption and poorer sleep health. After adjusting for confounders, females who did not consume caffeine reported more sleep disturbances (β = 0.64 [95%CI 0.13; 1.15]) compared to females who did consume caffeine (Table 2). In males, the direction of the association between caffeine consumption and sleep disturbances was similar to that in females but did not reach statistical significance. Females who did not consume caffeine had significantly higher odds of short sleep duration (OR = 2.26 [95% CI 1.22; 4.20]) compared to those who did consume caffeine (Table 3). No association was found between caffeine consumption and sleep duration in males. In both sexes, there was no statistically significant association between caffeine consumption and long sleep duration (Table 3), and no statistically significant association between caffeine consumption and perceived sleep quality (Table 4).

## 4. Discussion

Interaction between caffeine consumption and sex was statistically significant for each sleep outcome. Older females who did not consume caffeine reported more sleep disturbances and had greater odds of short sleep duration (<7 h/day) compared to those who did consume caffeine, whereas no associations were observed in males. In both sexes, after adjustment for confounders, there was no association between caffeine consumption and perceived sleep quality. Overall, a higher caffeine consumption was not associated with a poorer sleep health.

Although caffeine consumption has been linked to sleep disturbances, shorter sleep duration, and poorer perceived sleep quality in healthy adults [34], we did not observe this association in a representative sample of community-dwelling older males and females in The Netherlands. Our results are consistent with a previous observational study in middle-aged and older adults that found no association between self-reported caffeine consumption and sleep duration measured using actigraphy [38]. Most other studies evaluated caffeine intake from coffee consumption only [36,37,39], which may have led to a misclassification of subjects and an underestimation of associations. A large observational study (*n* = 8091) of adults aged 55 to 101 years in Europe did find an association between coffee consumption (i.e., drinking more than six cups of coffee per day) and shorter sleep duration [36]. Consumption of more than six cups of coffee per day corresponds to the ‘high’ to ‘very high’ caffeine consumption categories in the present study. Similarly, a cross-sectional study of 428 older Nigerian females found that habitual coffee consumption (yes versus no coffee) was associated with shorter sleep duration [37], contradicting our findings in older females. Furthermore, our observation that older females who abstained from caffeine consumption reported more sleep disturbances and shorter sleep duration than those who did consume caffeine is in line with the results of two observational studies of hospitalized older adults (59.3% female) [39] and community-dwelling older adults (43.1% female) [40]. In the latter study, caffeine consumption was measured using a questionnaire that assessed the intake of several caffeinated substances, and plasma caffeine concentrations were also measured in the late afternoon.

The results of our observational study contradict results from controlled experiments in laboratory settings, wherein the administration of ≥200 mg of caffeine resulted in prolonged sleep latency (difficulties falling asleep), reduced sleep efficiency (ratio of total sleep time to time in bed), and shorter sleep duration in young and middle-aged adults [35,41,42]. A key difference between controlled laboratory experiments and the present study is that caffeine exposure in the present study represents the participants’ habitual caffeine consumption. It has been hypothesized that habitual caffeine consumption may lead to the development of tolerance to caffeine, potentially reducing its effects on sleep in real-life circumstances [57,58]. A randomized experiment by Hindmarch et al. [59], for example, found that caffeine had a greater adverse effect on sleep health in adults with a lower habitual caffeine consumption compared to adults with a higher habitual caffeine consumption.

An alternative explanation could be that people with lower habitual caffeine consumption and people who abstain from caffeine consumption are more sensitive to the effects of caffeine and therefore consciously limit their caffeine consumption or adjust the timing of their consumption. This may explain the absence of an association between categories of caffeine consumption and sleep health in the present study. Indeed, there are significant interpersonal differences in caffeine sensitivity, which can be partially attributed to genetic factors. Studies have shown that certain genes—such as those involved in the metabolism of caffeine—may determine an individual’s physiological response to the stimulant as well as their consumption patterns [60,61]. The timing of caffeine consumption may impact the magnitude of its effect on sleep as the potential for sleep disturbance increases when caffeine is consumed closer to habitual bedtime [59,62]. Our observation that female older adults who abstained from caffeine consumption reported more sleep disturbances and shorter sleep duration compared to those who did consume caffeine may be explained by the possibility that females who experience sleep disturbances are aware of the stimulatory effects of caffeine and therefore refrain from its consumption. This presumed reverse causation and the observed sex differences therein are supported by evidence that females more often limit caffeinated coffee consumption than males or do so because they experience sleep disturbances [63]. Yet, our study suggests that sleep disturbances remain present despite refraining from caffeine consumption.

The present study has several strengths, including the use of a large, nationally representative sample of community-dwelling older adults, increasing the generalizability of the results. Additionally, detailed information on coffee and tea consumption was gathered to assess caffeine consumption (i.e., type of coffee/tea and cup volume). Furthermore, relevant potential confounding variables were accounted for in the analyses, although residual confounding cannot be fully disregarded. In addition to these strengths, this study has some limitations that need to be taken into account when interpreting the results. One limitation is that the timing of caffeine consumption was not taken into account. Another limitation is the reliance on self-reported measures of caffeine consumption and sleep health, which increases the likelihood of misclassification and bias due to subjectivity. Disparities in subjective and objective measurements of sleep have frequently been described. Observational evidence indicates that subjective report of habitual sleep duration may be biased by systematic overreporting, a consistent tendency of individuals to report longer sleep duration [64]. Discrepancies between subjective and objective findings may also result from discreet abnormalities in sleep quality that are difficult for individuals to notice but can be detected through the use of objective measurement tools such as polysomnography or actigraphy. Additionally, it has been suggested that individuals adjust their perception of what constitutes a good sleep as they age [65,66], making older adults generally more satisfied with their sleep, despite apparent disruptions in objective sleep quality [67,68,69]. Overall, the use of subjective measurements could potentially have led to random misclassification, resulting in an underestimation of the associations in the expected direction. Furthermore, the use of retrospective questions to assess coffee and tea consumption induces a potential for recall bias, especially given the higher prevalence of age-related cognitive and memory decline in older adults [70]. However, it is expected that coffee and tea consumption are well-recalled as habitual or frequently consumed foods are more accurately recalled than foods consumed less frequently or without a pattern [71,72]. Another limitation is the method used to calculate caffeine consumption from coffee and tea. Although a distinction was made between caffeinated and decaffeinated coffee, as well as the different types of tea, no information was available on the type of caffeinated coffee and tea, including the method of preparation, brand, or strength of the coffee or tea brew, which may determine the actual caffeine content [26]. Additionally, caffeine consumption was estimated based on coffee and tea consumption only, not taking into account caffeine intake through other dietary sources. However, this is unlikely to have led to a significant underestimation of caffeine consumption as coffee and tea are the primary sources of caffeine for adults in The Netherlands [27]. Lastly, this study is limited by the absence of a validated measurement instrument to assess sleep health. However, validated instruments such as the PSQI [73] and the Insomnia Severity Index [74] contain similar questions to assess sleep health; thus, we do not expect the results to be different.

The results of this study and previous research provide conflicting evidence. Therefore, the relationship between caffeine consumption and sleep health in older adults remains unclear. Given the high prevalence of sleep disturbances in older adults and the debilitating consequences associated with poor sleep health, further research is warranted. Epidemiological studies could help to better understand the relationship between caffeine and sleep in real-life settings. Such studies could consider the use of diaries to collect information on caffeine consumption and sleep health and may benefit from investigating the timing of caffeine consumption and the motivations behind older adults’ decisions to limit or abstain from caffeine consumption as this may confirm the potential reverse causation in older females. In addition, studies should consider the potential sex differences in the effects of caffeine on sleep as well as sex differences in caffeine-related behaviours.

## 5. Conclusions

Community-dwelling older females who did not consume caffeine reported more sleep disturbances and had higher odds of short sleep duration (<7 h/day) compared to females who did consume caffeine. In older males, no association was found between caffeine consumption and sleep health. The association found in older females may reflect reverse causation, suggesting that females may have different motivations for discontinuing caffeine use than males. In both sexes, there was no association between higher caffeine consumption and sleep health.

## Figures and Tables

**Figure 1 nutrients-16-00131-f001:**
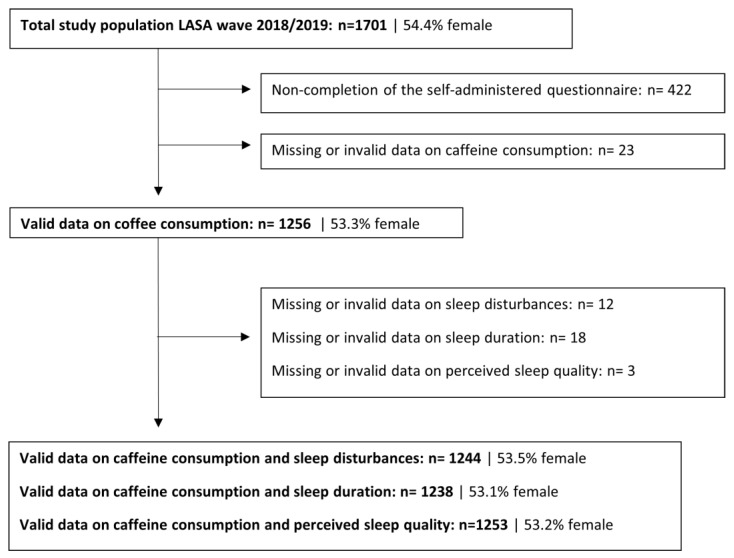
Flowchart of participants included in the analytic samples for the association between caffeine consumption from coffee and tea and sleep health in older adults from the Longitudinal Ageing Study Amsterdam (LASA).

**Table 1 nutrients-16-00131-t001:** Characteristics of the excluded and analytical sample (*n* = 1256), stratified by sex and by caffeine consumption (mg/day).

			Males			Females		
Characteristics	Excluded Cases (*n* = 455)	Analytical Sample (*n* = 1256)	Total (*n* = 587)	No Caffeine (*n* = 39)	Caffeine (*n* = 548)	Total (*n* = 669)	No Caffeine (*n* = 61)	Caffeine (*n* = 608)
**Caffeine consumption**								
Total caffeine (mg/day)								
Mean ± SD	-	263 ± 167	286 ± 175	-	306 ± 164	244 ± 156	-	268 ± 143
Categories of caffeine, N (%) ^a^								
None	-	100 (8.0)	39 (6.6)	39 (100)	-	61 (9.1)	61 (100)	-
Low	-	289 (23.0)	120 (20.4)	-	120 (21.9)	169 (25.3)	-	169 (27.8)
Moderate	-	289 (23.0)	126 (21.5)	-	126 (23.0)	163 (24.4)	-	163 (26.8)
High	-	300 (23.9)	153 (26.1)	-	152 (27.9)	147 (22.0)	-	147 (24.2)
Very high	-	278 (22.1)	149 (25.4)	-	149 (27.2)	129 (19.3)	-	129 (21.2)
Caffeine from coffee (mg/day) ^b^								
Median (IQR)	-	220 (194)	220 (187)	-	223 (220)	167 (220)	-	200 (182)
Caffeine from tea (mg/day) ^c^								
Median (IQR)	-	36 (102)	26 (75)	-	33.6 (78)	50 (116)	-	62 (123)
**Demographics**								
Age in years	*n* = 445							
Mean ± SD	76.3 ± 9.9	72.8 ± 7.5	72.5 ± 7.3	77.3 ± 10.2	72.2 ± 6.9	73.1 ± 7.7	75.9 ± 8.8	72.8 ± 7.5
Level of attained education, N (%)	*n* = 445							
Low	199 (44.7)	386 (30.7)	159 (27.1)	13 (33.3)	146 (26.6)	227 (33.9)	28 (45.9)	199 (32.7)
Intermediate	162 (36.4)	487 (38.8)	202 (34.4)	16 (41.0)	186 (33.9)	285 (42.6)	23 (37.7)	262 (43.1)
High	84 (18.9)	383 (30.5)	226 (38.5)	10 (25.6)	216 (39.4)	157 (23.5)	10 (16.4)	147 (24.2)
Partner status, N (%)	*n* = 443							
Living alone	212 (47.9)	408 (32.5)	125 (21.3)	8 (20.5)	117 (21.4)	283 (42.3)	29 (47.5)	254 (41.8)
Living with partner	231 (52.1)	848 (67.5)	462 (78.7)	31 (79.5)	431 (78.6)	386 (57.7)	32 (52.5)	354 (58.2)
Level of urbanization, N (%) *	*n* = 441							
Sparsely populated (<1000)	173 (39.2)	520 (41.5)	240 (41.0)	22 (56.4)	218 (39.9)	280 (41.9)	29 (47.5)	251 (41.4)
Densely populated (≥1000)	268 (60.8)	733 (58.5)	345 (59.0)	17 (43.6)	328 (60.1)	388 (58.1)	32 (52.5)	356 (58.6)
**Health related factors**								
Number of chronic diseases ^d^	*n* = 137							
Mean ± SD	2.8 ± 1.7	2.2 ± 1.4	2.1 ± 1.4	2.4 ± 1.6	2.1 ± 1.4	2.3 ± 1.4	2.5 ± 1.5	2.3 ± 1.4
Depressive symptoms (CES-D) *	*n* = 137							
Median (IQR)	10 (12)	5.0 (8.0)	5.0 (7.0)	5.0 (6.0)	4.0 (7.0)	6.0 (8.0)	8.0 (10)	6.0 (8.0)
Body mass index (kg/m^2^) *	*n* = 103							
Mean ± SD	27.2 ± 4.7	27.3 ± 4.7	27.4 ± 4.2	27.0 ± 5.2	27.5 ± 4.1	27.2 ± 5.2	28.3 ± 5.8	27.1 ± 5.1
Subjective pain *								
Mean ± SD	-	5.8 ± 1.4	5.6 ± 1.2	5.8 ± 1.2	5.6 ± 1.2	6.0 ± 1.6	6.4 ± 1.9	6.0 ± 1.5
**Behavioural factors**								
Smoking status, N (%) *	*n* = 107							
Never smoked	26 (24.3)	304 (24.9)	116 (20.3)	7 (18.4)	109 (20.5)	188 (28.9)	20 (33.3)	168 (28.5)
Former smoker	66 (61.7)	812 (66.5)	410 (71.8)	30 (78.9)	380 (71.3)	402 (61.8)	35 (58.3)	367 (62.6)
Current smoker	15 (14.0)	105 (8.6)	45 (7.9)	1 (2.6)	44 (8.3)	60 (9.2)	5 (8.3)	55 (9.3)
Alcohol use, N (%) *	*n* = 107							
No use	35 (32.7)	172 (14.1)	56 (9.8)	7 (18.4)	49 (9.2)	116 (17.8)	19 (31.7)	97 (16.4)
Moderate use	67 (62.6)	925 (75.8)	467 (81.8)	31 (81.6)	436 (81.8)	458 (70.5)	38 (63.3)	420 (71.2)
Above moderate use	5 (4.7)	124 (10.2)	48 (8.4)	-	48 (9.0)	76 (11.7)	3 (5.0)	73 (12.4)
Physical activity (MET-hrs./wk.) *	*n* = 136							
Median (IQR)	35.7 (49.1)	52.0 (45.3)	45.2 (40.6)	37.3 (43.2)	45.6 (40.2)	58.6 (45.2)	51.0 (41.1)	59.3 (45.5)
**Sleep parameters**								
Sleep duration(h/day), N (%) *								
Mean ± SD	-	7.4 ± 1.1	7.5 ± 1.1	7.8 ± 1.0	7.5 ± 1.1	7.3 ± 1.1	6.9 ± 1.3	7.3 ± 1.1
Recommended (7–8 h)	-	795 (64.2)	402 (69.2)	27 (71.1)	375 (69.1)	393 (59.8)	24 (42.9)	369 (61.4)
Short (< 7 h)	-	272 (22.0)	99 (17.0)	5 (13.2)	94 (17.3)	173 (26.3)	25 (44.6)	148 (24.6)
Long (>8 h)	-	171 (13.8)	80 (13.8)	6 (15.8)	74 (13.6)	91 (13.9)		7 (12.5) 84 (14.0)
Sleep disturbances, N (%) ^e^ *								
Median (IQR)	-	6.0 (3.0)	5.0 (3.0)	6.0 (2.0)	5.0 (3.0)	6.0 (3.0)	7.0 (3.0)	6.0 (3.0)
No disturbances (3)	-	154 (12.4)	101 (17.4)	2 (5.1)	99 (18.3)	53 (8.0)	2 (3.3)	51 (8.4)
Some disturbances (4–6)	-	652 (52.4)	329 (56.8)	25 (64.1)	304 (56.3)	323 (48.6)	23 (37.7)	300 (49.7)
Several disturbances (7–9)	-	376 (30.2)	131 (22.6)	12 (30.8)	119 (22.0)	245 (36.8)	29 (47.5)	216 (35.8)
Many disturbances (10–12)	-	62 (5.0)	18 (3.1)	-	18 (3.3)	44 (6.6)	7 (11.5)	37 (6.1)
Perceived sleep quality, N (%) *								
Good	-	1028 (82.0)	511 (87.2)	36 (92.3)	475 (86.8)	517 (77.5)	44 (72.1)	473 (78.1)
Poor	-	225 (18.0)	75 (12.8)	3 (7.7)	72 (13.2)	150 (22.5)	17 (27.9)	133 (21.9)

*n* = number of participants, SD = standard deviation, IQR = interquartile range. Sample characteristics are based on non-imputed data. Variables with missing data (<5%) are denoted with an asterisk (*). ^a^ None: 0 mg/day, low: ≤ 173 mg/day, moderate: 174–272 mg/day, high: 273–367 mg/day, and very high: >367 mg/day, ^b^
*n* = 1223, ^c^
*n* = 1128, ^d^. Including nonspecific lung disease, cardiac disease, peripheral arterial disease, stroke, diabetes mellitus, rheumatoid arthritis, and malignancies as well as up to two other self-reported chronic diseases, ^e^ Measured on a 3 to 12 scale based on three questions that assessed the frequency of experiencing difficulties with sleep latency, continuity, and early morning awakening. Higher scores indicate more sleep disturbances.

**Table 2 nutrients-16-00131-t002:** Crude and adjusted linear regression models for the association between caffeine consumption (mg/day) and sleep disturbances (measured on a 3–12 scale ^b^), stratified by sex (*n* = 1244).

	Males (*n* = 579)			Females (*n* = 665)		
	β	95% CI	*p*-Value	β	95% CI	*p*-Value
**Categories of caffeine ^a^**						
Crude model						
None	0.12	−0.58–0.82	0.737	0.80	0.20–1.40	**0.009**
Low (ref.)	-	-	-	-	-	-
Moderate	−0.27	−0.76–0.21	0.267	0.06	−0.38–0.51	0.776
High	−0.54	−1.00–−0.07	**0.024**	−0.23	−0.68–0.23	0.326
Very high	−0.21	−0.68–0.26	0.382	−0.14	−0.61–0.33	0.561
Adjusted model ^c^						
None	0.23	−0.44–0.90	0.500	0.61	0.04–1.17	**0.035**
Low (ref.)	-	-	-	-	-	-
Moderate	−0.17	−0.63–0.29	0.467	0.08	−0.34–0.50	0.713
High	−0.35	−0.80–0.09	0.122	−0.12	−0.55–0.31	0.572
Very high	−0.12	−0.58–0.33	0.597	−0.12	−0.57–0.32	0.587
**Caffeine vs. no caffeine**						
Crude model						
Caffeine (ref.)	-	-	-	-	-	-
No caffeine	0.39	−0.24–1.02	0.222	0.87	0.33–1.41	**0.002**
Adjusted model ^c^						
Caffeine (ref.)	-	-	-	-	-	-
No caffeine	0.39	−0.21–1.00	0.205	0.64	0.13–1.15	**0.014**

β = beta coefficient, CI = confidence interval, ref. = reference category. ^a^ None: 0 mg/day, low: ≤173 mg/day, moderate: 174–272 mg/day, high: 273–367 mg/day, and very high: >367 mg/day, ^b^ Higher scores indicate more sleep disturbances, ^c^ Adjusted for age, level of attained education, partner status, level of urbanization, number of chronic diseases, depressive symptoms (CES-D), BMI (kg/m^2^), subjective pain, smoking status, alcohol use, and physical activity (MET-hrs./wk.).

**Table 3 nutrients-16-00131-t003:** Crude and adjusted multinomial logistic regression models for the association between caffeine consumption (mg/day) and sleep duration, stratified by sex (*n* = 1238).

	Males (*n* = 581)			Females (*n* = 657)		
	OR ^b^	95% CI	*p*-Value	OR	95% CI	*p*-Value
**Short sleep duration (<7 h ) ^a^**						
**Categories of caffeine ^c^**						
Crude model						
None	0.78	0.27–2.29	0.651	2.45	1.26–4.75	**0.008**
Low (ref.)	-	-	-	-	-	-
Moderate	0.91	0.45–1.84	0.791	0.93	0.55–1.55	0.767
High	1.20	0.63–2.29	0.587	1.00	0.59–1.69	0.997
Very high	1.08	0.56–2.08	0.812	0.83	0.48–1.45	0.511
Adjusted model ^d^						
None	0.93	0.30–2.87	0.903	2.18	1.09–4.37	**0.028**
Low (ref.)	-	-	-	-	-	-
Moderate	0.94	0.45–1.96	0.867	0.97	0.56–1.66	0.899
High	1.24	0.62–2.47	0.537	1.10	0.63–1.91	0.744
Very high	1.00	0.49–2.02	0.989	0.80	0.45–1.44	0.454
**Caffeine vs. no caffeine**						
Crude model						
Caffeine (ref.)	-	-	-	-	-	-
No caffeine	0.74	0.28–1.97	0.545	2.60	1.44–4.69	**0.002**
Adjusted model ^d^						
Caffeine (ref.)	-	-	-	-	-	-
No caffeine	0.90	0.32–2.48	0.833	2.26	1.22–4.20	**0.010**
**Long sleep duration (>8 h) ^a^**						
**Categories of caffeine ^c^**						
Crude model						
None	0.94	0.34–2.59	0.898	1.28	0.49–3.33	0.612
Low (ref.)	-	-	-	-	-	-
Moderate	0.81	0.40–1.67	0.574	1.07	0.56–2.01	0.847
High	0.91	0.46–1.79	0.781	1.06	0.55–2.05	0.862
Very high	0.64	0.31–1.33	0.231	0.86	0.43–1.73	0.666
Adjusted model ^d^						
None	0.73	0.24–2.17	0.564	1.19	0.44–3.18	0.735
Low (ref.)	-	-	-	-	-	-
Moderate	0.89	0.42–1.90	0.761	1.08	0.55–2.10	0.827
High	1.19	0.57–2.46	0.646	1.17	0.59–2.34	0.657
Very high	0.82	0.38–1.78	0.622	0.97	0.46–2.02	0.925
**Caffeine vs. no caffeine**						
Crude model						
Caffeine (ref.)	-	-	-	-	-	-
No caffeine	1.13	0.45–2.82	0.800	1.28	0.53–3.07	0.579
Adjusted model ^d^						
Caffeine (ref.)	-	-	-	-	-	-
No caffeine	0.75	0.28–2.03	0.567	1.13	0.46–2.80	0.788

OR = odds ratio, CI = confidence interval, ref. = reference category. ^a^ The reference category is the recommended sleep duration (7–8 h/day). ^b^ An OR > 1 indicates higher odds of short or long sleep duration, compared to the recommended sleep duration. ^c^ None: 0 mg/day, low: ≤173 mg/day, moderate: 174–272 mg/day, high: 273–367 mg/day, and very high: >367 mg/day, ^d^ Adjusted for age, level of attained education, partner status, level of urbanization, number of chronic diseases, depressive symptoms (CES-D), BMI (kg/m^2^), subjective pain, smoking status, alcohol use, and physical activity (MET-hrs/wk.).

**Table 4 nutrients-16-00131-t004:** Crude and adjusted logistic regression models for the association between caffeine consumption (mg/day) and perceived sleep quality, stratified by sex (*n* = 1253).

	Males (*n* = 586)			Females (*n* = 667)		
	OR ^a^	95% CI	*p*-Value	OR	95% CI	*p*-Value
**Categories of caffeine ^b^**						
Crude model						
None	0.58	0.16–2.13	0.415	1.42	0.73–2.77	0.308
Low (ref.)	-	-	-	-	-	-
Moderate	0.96	0.45–2.05	0.905	1.09	0.65–1.83	0.758
High	0.82	0.39–1.73	0.598	0.90	0.52–1.56	0.710
Very high	1.48	0.75–2.94	0.263	1.16	0.67–2.00	0.595
Adjusted model ^c^						
None	0.86	0.22–3.47	0.837	1.25	0.60–2.58	0.556
Low (ref.)	-	-	-	-	-	-
Moderate	1.18	0.50–2.77	0.711	1.04	0.59–1.82	0.903
High	1.03	0.45–2.40	0.938	0.97	0.54–1.76	0.917
Very high	1.97	0.89–4.39	0.095	1.11	0.61–2.02	0.738
**Caffeine vs. no caffeine**						
Crude model						
Caffeine (ref.)	-	-	-	-	-	-
No caffeine	0.55	0.17–1.83	0.330	1.37	0.76–2.48	0.293
Adjusted model ^c^						
Caffeine (ref.)	-	-	-	-	-	-
No caffeine	0.67	0.19–2.41	0.543	1.21	0.64–2.32	0.558

OR = odds ratio, CI = confidence interval, ref. = reference category. ^a^ An OR > 1 indicates higher odds of poor perceived sleep quality. ^b^ None: 0 mg/day, low: ≤173 mg/day, moderate: 174–272 mg/day, high: 273–367 mg/day, and very high: >367 mg/day, ^c^ Adjusted for age, level of attained education, partner status, level of urbanization, number of chronic diseases, depressive symptoms (CES-D), BMI (kg/m^2^), subjective pain, smoking status, alcohol use, and physical activity (MET-hrs/wk.).

## Data Availability

The data presented in this study are available on request from the corresponding author. The data are not publicly available due to privacy restrictions.

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
