# Peer review of "The Association between Caffeine Consumption from Coffee and Tea and Sleep Health in Male and Female Older Adults: A Cross-Sectional Study"

_nutrients, 2023, doi:10.3390/nu16010131_

Round 1

Reviewer 1 Report

Comments and Suggestions for Authors

Review:  The Association between Caffeine Consumption from Coffee 2 and Tea and Sleep Health in Male and Female Older Adults: 3 A Cross-Sectional Study

The article is very interesting and well written. The authors make a good synthesis of the available information, and description of the sample. I add some comments for the authors to consider.

2.1. Design and Participants

The approval code of the study should be displayed. It should also indicate whether all participants were volunteers and signed the informed consent form.

2.2. Sleep Disturbances, Sleep Duration, and Perceived Sleep Quality

The questions used to measure sleep “Do you experience difficulties falling asleep?’, ‘Do 107 you experience interruptions in your sleep?’, and ‘Do you wake up too early?’. Response 108 options to each of the three questions were: ‘almost never’, ‘sometimes’, ‘frequently’, and 109 ‘almost always’.” + ““When you think 118 about the past month, how would you rate your sleep quality?” Are they validated? The authors should indicate this.

2.3. Caffeine Consumption from Coffee and Tea

Was the consumption of these foods collected with a validated food frequency questionnaire?

2.4. Other Measurements

Is there information on the presence of chronic cardiovascular diseases, diabetes or cholesterol? These variables could be taken into account in model fitting. Instead of "no. of chronic diseases".

Is information on calorie intake available? This should also be taken into account.

2.5. Statistical Analyses

The authors have forgotten to add how they represent the categorical variables in the tables.

Have you considered stratifying the analyses by caffeine consumption according to whether it is coffee or tea?

In addition, a stratified analysis by age group should be carried out, as more sleep problems have been observed at older ages.

3. Results

Authors can remove the SD from the text and the IQR in the description of the results (lines 235-246) to make it easier to read.

The confidence interval for the low consumption category is very wide, and not very stable. Perhaps due to the small sample size of this category (39 and 61). It might have been appropriate to combine the non-consumption category with low consumption.

Discussion

The authors have already outlined the objectives of the study. Start the discussion by highlighting their most important results. Delete lines 299-302

The authors discuss their results with studies that have evaluated caffeine intake from coffee consumption. Perhaps they should bear in mind that their results are from coffee and tea blends. This should be taken into account.

Author Response

Dear reviewer,

Please find the response in the attached document. 

With kind regards,

Mette van der Linden

Reviewer 2 Report

Comments and Suggestions for Authors

Dear Authors,

Next to tea, coffee is one of the most popular drinks in the world. Opinions on coffee are very contradictory. There are plenty of studies that talk about the positives of drinking coffee, just as there are those that point out its downsides. But what is the relationship between caffeine and sleep really like? The authors of the article investigated how caffeine affects older people from the Netherlands. In doing so, they demonstrated that the effect of caffeine on individuals can vary according to gender, age and many other factors.

In my opinion:

§  The manuscript uses a clear introduction;

§  The material and research methods were described in detail;

§  The research was properly planned and collected;

§  Proper statistical methods were used.

Please answer my concerns:

1.      How does the caffeine in coffee interfere with melatonin secretion?

2.      What are the literature data answering the question at what time to drink coffee to avoid sleep problems?

3.      Why can caffeine be classified as a psychoactive substance?

Author Response

(The authors gave the same response as above.)

Reviewer 3 Report

Comments and Suggestions for Authors

The manuscript entitled “The Association between Caffeine Consumption from Coffee and Tea and Sleep Health in Male and Female Older Adults: A Cross-Sectional Study” by Linden et al is informative study. The study has addressed the association between caffeine consumption and sleep health in older people.  However, following concerns need to be addressed and reconciled which could improve/upgrade this manuscript.

·       Older aged major population already shows lack of sleep. State the major reason to choose this age group for the study. 

·       Authors should write about how many hours prior to habitual bedtime they consume caffeine.

·       The authors should specify the inclusion and exclusion criteria for the study sample in the materials and methods section, and experimental design must be depicted in flow chart format

·       In section 2.2, authors must include given criteria with number of participants and gender in flowchart format.

·       Authors must mention the number of males and females who were encountered with the questions of caffeine consumption

·       Line 134-157, the calculation is very confusing in written paragraph format, authors should correct it and must be included in a separate column of methodology flow chart

·       How an attained education and partner status related to caffeine consumption and how these measurements are important for this study

·       Did authors also check the status of financial problems and addicted people in this study?

·       Conclusion must include the importance of the study as well and future prospects

·       Author must read the manuscript very carefully and correct the language and grammatical error of the of whole the manuscript (line no. 19, 111, 123, 128, 303 etc).

·       I suggest to the authors to show his notably important finings data in graphical format to enhance the visibility of their manuscript.

Thanks

Author Response

(The authors gave the same response as above.)
